# Factors Influencing the Outcome of Head and Neck Cancer of Unknown Primary (HNCUP)

**DOI:** 10.3390/jcm11102689

**Published:** 2022-05-10

**Authors:** Matthias Balk, Robin Rupp, Konstantinos Mantsopoulos, Matti Sievert, Magdalena Gostian, Moritz Allner, Philipp Grundtner, Markus Eckstein, Heinrich Iro, Markus Hecht, Antoniu-Oreste Gostian

**Affiliations:** 1Department of Otolaryngology, Head & Neck Surgery, Comprehensive Cancer Center Erlangen-Europäische Metropolregion Nürnberg, University Hospital Erlangen, Friedrich-Alexander University Erlangen-Nürnberg (FAU), Waldstraße 1, 91054 Erlangen, Germany; robin.rupp@uk-erlangen.de (R.R.); konstantinos.mantsopoulos@uk-erlangen.de (K.M.); matti.sievert@uk-erlangen.de (M.S.); moritz.allner@uk-erlangen.de (M.A.); philipp.grundtner@uk-erlangen.de (P.G.); heinrich.iro@uk-erlangen.de (H.I.); antoniu-oreste.gostian@uk-erlangen.de (A.-O.G.); 2Department of Anaesthesiology and Intensive Care Medicine, Malteser Waldkrankenhaus St. Marien, 91054 Erlangen, Germany; magdalenaf@gmx.de; 3Department of Pathology, Comprehensive Cancer Center Erlangen-Europäische Metropolregion Nürnberg, University Hospital Erlangen, Friedrich-Alexander University Erlangen-Nürnberg (FAU), Krankenhausstraße 8-10, 91054 Erlangen, Germany; markus.eckstein@uk-erlangen.de; 4Department of Radiation Oncology, Comprehensive Cancer Center Erlangen-Europäische Metropolregion Nürnberg, University Hospital Erlangen, Friedrich-Alexander University Erlangen-Nürnberg (FAU), Universitätsstraße 27, 91054 Erlangen, Germany; markus.hecht@uk-erlangen.de

**Keywords:** CUP syndrome, head and neck cancer, neck dissection, multivariate analysis, survival

## Abstract

Background: This study on patients with head and neck cancer of unknown primary (HNCUP) assesses the impact of surgical and non-surgical treatment modalities and the tumour biology on the oncological outcome. Methods: A total of 80 patients with HNCUP (UICC I–IV) were treated with simultaneous neck dissection followed by adjuvant therapy, between 1 January 2007 and 31 March 2020. As the primary objective, the influence of treatment modalities on the overall survival (OS), the disease-specific survival (DSS) and the progression-free survival (PFS) were analysed in terms of cox regression and recursive partitioning. The tumour biology served as secondary objectives. Results: The 5-year OS for the entire cohort was 67.7%, (95% CI: 54.2–81.2%), the 5-year DSS was 82.3% (72.1–92.5%) and the 5-year PFS was 72.8% (61.8–83.8%). Cox regression revealed that patients undergoing adjuvant radiotherapy only had a four times higher risk to die compared to patients receiving chemoradiation therapy (HR = 4.45 (1.40; 14.17), *p* = 0.012). The development of distant metastases had a significantly negative impact on OS (HR = 8.24 (3.21–21.15), *p* < 0.001) and DSS (HR = 23.79 (6.32–89.56), *p* < 0.001). Recursive portioning underlined the negative influence of distant metastases on OS (3.2-fold increase in death probability) and DSS (4.3-fold increase in death probability), while an UICC stage of IVb increased the risk for further progression of the disease by a factor of 2. Conclusions: The presence of distant metastases as well as adjuvant treatment with radiation without concomitant chemotherapy, were among others, significant predictors for the overall survival of HNCUP patients, with distant metastases being the most significant predictor.

## 1. Introduction

The diagnosis of “cancer of unknown primary” describes a metastasis whose primary tumour location cannot be found, and which can occur in different areas of the human body. It has been found that 2–5% of head and neck cancer patients present with a cervical lymph node metastasis without a detectable primary tumour, leading to the diagnosis of head and neck cancer of unknown primary (HNCUP) [1,2,3].

Among many entities, the most common histology is squamous cell carcinoma, which affects 53–77% of patients [4]. In this context, the differentiation between human papilloma virus (HPV)-positive and -negative HNCUP syndromes has been shown to distinctly affect survival with favourable outcomes for HPV-positive HNCUP [5]. Reported HPV prevalence varies from 22% to 91% [2,5,6,7,8], which is reflected in 5-year survival rates that range from 29% to 82% [2,5,6,7,8,9].

In advanced-stage disease, i.e., UICC tumour stages III and IV, the combined therapeutical approach consisting of neck dissection followed by adjuvant radiation/chemoradiation is highly recommended as it yields the most favourable oncological outcomes [10,11,12,13,14]. In contrast, for early stage disease, i.e., UICC tumour stages I and II, the most effective therapy, i.e., ipsilateral selective neck dissection followed by radiation therapy or chemoradiation and primary radiation/chemoradiation therapy, is still under debate [15].

The aim of the study presented here is to evaluate the results achieved by neck dissection followed by adjuvant therapy in patients with HNSCC-CUP. Accordingly, the primary objective focusses on the impact of the treatment characteristics on the oncological outcome demonstrated by 5-year overall survival, 5-year disease-specific survival and 5-year progression-free survival. Secondary objectives included, among others, the nodal stage, presence of distant metastases, HPV status and extranodular extension (ENE).

## 2. Materials and Methods

The following retrospective cohort study was conducted at a single tertiary referral and academic cancer centre. It was approved by the local Ethics Committee (approval number 428_20 Bc) and carried out in accordance with the Declaration of Helsinki.

All patients diagnosed with an HNCUP syndrome between 1 January 2007 and 31 March 2020 were included. The following inclusion criteria applied: surgical treatment at our institution, adjuvant therapy, histologically confirmed squamous cell carcinoma, complete medical and surgical record available and evaluated p16 status. The association with HPV was confirmed by overexpression of the surrogate marker p16INK4a.

The following exclusion criteria applied: histological types other than squamous cell carcinoma, detection of the primary tumour during the diagnostic workup, unknown or absent p16 status or discontinued treatment.

Alcohol consumption was defined as the current daily alcohol consumption reported. Smoking was defined as current smokers with a smoking history of at least 10 pack years.

The treatment process for HNCUPs involved first of all a “no-touch” panendoscopy, followed either by a core needle biopsy or node picking of the suspicious lymph node. Afterwards, a positron emission tomography (PET) was performed. If the primary cancer was not detected, the PET would be followed by a CUP panendoscopy that included a curettage of the nasopharynx, multiple biopsies of the tongue base and a bilateral tonsillectomy. Simultaneously, an adapted neck dissection was performed on the side of the previously diagnosed malignant lymph node. A bilateral neck dissection was performed in the case of clinically suspicious contralateral lymph nodes. 

In the case of ENE and more than one affected lymph node, adjuvant treatment was applied as chemoradiation therapy, and in cases with only one affected lymph node only radiation therapy was performed [16].

The neck dissections performed were classified according to Robbins et al. Selective neck dissection indicates the preservation of one or more groups of lymph nodes, modified radical neck dissection indicates the preservation of one or more non-lymphatic structures and radical neck dissection indicates the removal of the internal jugular vein, the spinal accessory nerve and the sternocleidomastoid muscle besides the lymph node groups [17]. In addition, the lymph node ratio (LNR) was defined as the number of positive lymph nodes divided by the total number of lymph nodes removed [18,19].

Radiation techniques included volumetric modulated arc therapy (VMAT), 3D conformal radiation therapy or intensity-modulated radiation therapy (IMRT). The radiation dose was 56 Gray (Gy) in the ipsilateral neck and 64 Gy in the area of the affected lymph node. Possible primary sites (i.e., hypopharynx, tongue base and tonsils bilaterally) and the contralateral neck received 50 Gy, whereas 70 Gy were delivered to the nasopharynx based on the institution’s own guidelines. Radiation was applied either as a simultaneous, integrated boost using single doses of up to 2.3 Gy to biologically equivalent cumulative doses or in single doses of 2.0 Gy sequentially in the shrinking field technique. Concomitant chemotherapy consisted of two cycles of cisplatin (100 mg/m^2^ BSA) or carboplatin (AUC 5) in combination with 5-Fluorouracil (800 mg/m^2^ body surface area (BSA) continuous infusion d1–5) split into 3–5 days. Three patients were not suitable for cisplatin/carboplatin, and in this case the patients received cetuximab (400 mg/m^2^ BSA once a week).

### 2.1. Study Objectives

The 5-year overall survival rate was calculated from the date of the neck dissection to the date of death from any cause. The 5-year disease-specific survival rate was calculated from the date of the neck dissection to the date of death from the disease, and the 5-year progression-free survival rate was calculated from the date of the neck dissection to the date of progression of the disease. Patients who were lost to follow-up and those who were still alive at the time of the follow-up cut-off were censored. Recurrence of disease was defined as regional tumour recurrence or distant metastasis. Follow-up consisted of a clinical examination performed by an ENT specialist every 6 weeks in the first year, every three months in the second and third years and every six months in the fourth and fifth years after completion of initial treatment. A thorough ultrasound covering both sides of the neck was performed at each follow-up appointment and a computed tomography scan of the neck and thorax was performed once a year. The tumour stage was classified according to the 8th version of the UICC [20].

### 2.2. Statistical Analysis

Continuous variables were tested for normal distribution using the Kolmogorov–Smirnov tests, the histogram and a QQ plot. Based on the result, they are presented as mean ± 1 standard deviation or median (25th and 75th percentile). Nominal variables are presented as absolute and relative frequencies (N/%). Overall survival, disease-specific survival and progression-free survival were calculated using Kaplan–Meier curves; the 5-year OS/DSS/PFS was presented including its 95% confidence interval (OS/DSS/PFS (95% CI)).

The identification of risk factors for OS/DSS/PFS was conducted using Cox regressions. For this, a univariate Cox regression was performed on each predictor first, followed by a multiple Cox regression for those predictors identified as significant risk factors in the univariate analysis. A stepwise backward algorithm was chosen with *p* = 0.07 as exclusion threshold. For each tested predictor, the hazard ratio including its 95% confidence interval (HR (95%-CI)), Wald statistic, and descriptive statistics in terms of N/% or mean ± 1 SD are reported. 

In addition, we conducted survival tree analysis using recursive partitioning with the *rpart* package in R [21]. The goal of this analysis is to explain the survival of the patient population using the smallest possible number of predictors/splits. To this end, decisions between competing models are made based on the “min (xerror) + xstd” or 1-SE rule. This means that first, the group of models that fall within the range of the minimum cross-validation error (plus its standard error) are considered. Then, second, the model with the fewest splits among this group is chosen. Thus, the rule avoids overfitting.

Statistical analyses were performed in SPSS and R.

## 3. Results

### 3.1. Patient Characteristics

A total of 131 patients presented with a HNCUP syndrome at our department between 1 January 2007, and 31 March 2020. Out of these, 80 patients were eligible for the study and were consequently included in the analysis. The entire patient cohort averaged 60.6 years (yrs) (SD = 10.3 years) and included 66 male patients (82.5%).

While a total of 59 patients (73.8%) had a negative profile for the surrogate marker p16INK4a, 21 patients (26.2%) had a positive p16 status and were diagnosed with HPV-positive HNCUP.

The majority of patients were diagnosed at advanced stages according to UICC IVA (n = 27; 33.8%) and IVB (n = 28; 35%). Moreover, 40 out of 80 patients (50%) had a pN3b nodal stage. The patient characteristics are presented in Table 1.

### 3.2. Treatment Characteristics

All patients received an adapted neck dissection followed by adjuvant therapy. The neck dissection was performed in 33 patients on the left side (41.2%), in 43 patients on the right side (53.8%) and in 4 patients simultaneously on both sides (5%). Furthermore, 34 patients had less than 18 identified lymph nodes (42.5%).

In total, 43 patients (53.8%) received selective neck dissection, 14 patients (17.5%) modified radical neck dissection and 23 patients (28.7%) underwent radical neck dissection. A total of 72 patients (90%) received adjuvant chemoradiation, in contrast to 8 patients (10%) who underwent adjuvant radiation only. The mean follow-up time was 44.1 months (SD = 35.9 months). Table 2 shows the treatment modalities applied.

### 3.3. Oncological Outcomes

The 5-year overall survival rate of the entire cohort was 67.7%, 95% CI (54.2–81.2%). The 5-year disease-specific survival and the 5-year progression-free survival were 82.3% (72.1–92.5%) and 72.8% (61.8–83.8%), respectively. Six patients (7.5%) experienced a regional recurrence after a median of 12 months (1.5; 56.3) and 13 patients (16.3%) developed distant metastases after a median of 11 months (9; 16.5). Table 3 shows the oncological outcomes.

The univariate Cox regression of OS, in which all predictors were first tested individually, revealed that patients who received modified radical neck dissection had a 3.77-fold increased risk of death compared with those who received selective neck dissection (HR = 3.77 (1.25–11.40), *p* = 0.019). There was no significant difference between patients with radical neck dissection and selective neck dissection (HR = 2.24 (0.83–6.04), *p* = 0.110).

Patients who received adjuvant radiation therapy alone had a 3.54-fold increased risk of death compared with those who received adjuvant chemoradiation (HR = 3.54 (1.16–10.81), *p* = 0.027). Patients who presented with distant metastases during the follow-up had a 7.41-fold increased risk of dying (HR = 7.41 (2.95–18.62), *p* < 0.001).

In the multiple Cox regression, except for the type of neck dissection, all significant predictors of the univariate Cox regression mentioned above retained their significance (radiotherapy vs. chemoradiation therapy: HR = 4.45 (1.40–14.17), *p* = 0.012; distant metastases HR = 8.24 (3.21–21.15), *p* < 0.001). Table 4 shows the univariate and multiple Cox regression for OS.

Recursive partitioning revealed a decreased overall survival in patients with distant metastases: 9 out of 13 (69%) patients with distant metastases died until the end of observation compared to 13 out of 67 (19.4%) in patients without distant metastases. This resulted in a 3.2-fold increase regarding the risk of dying compared to the overall population. Figure 1 displays the survival curves of patients with and without distant metastases.

Regarding DSS, the univariate Cox regression revealed that patients who received selective neck dissection were 5.35 times more likely to have better DSS compared to those who received modified radical neck dissection (HR = 5.35 (1.06–27.09), *p* = 0.043). Patients with radical neck dissection had a 4.52-fold increased risk compared with those who received selective neck dissection (HR = 4.52 (1.23–18.14), *p* = 0.033).

Patients with locoregional metastases had a 4.89-fold increased risk of dying (HR = 4.89 (1.03–23.15), *p* = 0.046), and patients with distant metastasis had a 23.79-fold increased risk (HR = 23.79 (6.32–89.56), *p* < 0.001).

Patients with a higher LNR also had a higher risk of dying for disease-specific reasons, i.e., the fewer lymph nodes harvested per affected lymph node, the more likely the patients were to die from disease-specific causes (HR = 12.93 (1.54–108.45), *p* = 0.018). 

In the multiple Cox regression, only the presence of distant metastases remained as significant predictor: patients with distant metastases had a 23.79-fold increased risk of dying from disease-specific causes than those without (HR = 23.79 (6.32–89.56), *p* < 0.001). Table 5 shows the univariate and multiple Cox regression for DSS.

Recursive partitioning revealed that distant metastases resulted in a decreased probability to survive for disease specific reasons, with 9 out of 13 (69.2%) patients with distant metastases dying over time compared to only 3 out of 67 (4.5%) without distant metastases. This resulted in a 4.3-fold increase regarding the risk to die in patient with distant metastases compared to the general population. Figure 2 displays the disease specific survival curves of patients with and without distant metastases.

In the univariate Cox regression of PFS, ENE increased the risk of locoregional recurrence 8.69-fold (HR = 8.69 (1.16–65.37), *p* = 0.036). Performance of a modified radical neck dissection increased the risk 5.42-fold compared with selective neck dissection (HR = 5.42 (1.69–17.40), *p* = 0.004).

A higher LNR proved to be a significant unfavourable factor (HR = 7.93 (1.38–45.45), *p* = 0.020) also for PFS. 

Distant and locoregional metastases were removed from this Cox regression, as both together represent the status to be predicted.

Multiple Cox regression showed that patients with extranodal extension had an 8.13-fold increased risk of recurrence (HR = 8.13 (1.08–61.34), *p* = 0.042). In addition, a higher LNR was significantly associated with a distinctly increased risk of developing locoregional recurrence (HR = 6.17 (1.13–33.76), *p* = 0.036). Table 6 shows the univariate and multiple Cox regression for PFS.

Recursive partitioning revealed that an UICC stage of II or IVb resulted in a decreased probability to survive without further progression of the disease. While 14 out of 31 (35.9%) with UICC stage II and IVb had a progress of any kind during observation time, only 5 out 49 with UICC stage I, III, and IVa (10.2%) experienced a progression until the end of observation. This resulted in a two-fold increase in the risk of having a clinical progression for patients with UICC stage II and IVb compared to the general population.

It must be noted, though that only 3 patients of the clinical population had UICC stage II, so the estimation of the effect may unprecise. Figure 3 displays the disease specific survival curves of patients with different UICC stages.

## 4. Discussion

The results of the study presented here, showing a 5-year OS of 67.7%, a 5-year DSS of 82.3% and a 5-year PFS of 72.8%, are comparable to reported outcomes of HNCUP that range from 40.9% to 78.9% [4,22,23,24,25]. It is of note that, in comparison to older studies exemplified by the evaluation of Grau et al. from the years 1975 to 1995 with an OS of 37% [1], the improvement in the treatment of HNCUP over the decades is evident.

The thorough analysis of the oncological results with the multiple and survival tree analysis showed that the development of distant metastases had a negative impact on OS and DSS. Furthermore, the adjuvant treatment modality was identified as having a distinct impact on OS. The difference between radiation alone compared to chemoradiation therapy showed, in the multiple analysis, a significant disadvantage for OS in the patients who received adjuvant radiation only. However, it should be noted that in our cohort, only 8 patients received radiation therapy alone. Three patients refused chemotherapy, two discontinued it due to side effects and three others received the tumour board recommendation of radiation therapy alone based on the pN1 status. Remarkably, RT had a clear impact on OS but not on PFS and DSS, indicating the fact that these patients died for reasons other than tumour progression.

Considering the individual tumour characteristics, a significant, negative influence on OS, DSS and PFS was shown. Additionally, multiple analyses displayed that a higher LNR had a significantly negative impact on PFS, while the presence of ENE influenced PFS negatively.

The survival tree analysis also displayed that UICC stage II and IVb have a negative impact on PFS. However, stage II must be viewed critically, as only three patients were included in the study presented. After all, the fact that a stage IVb has a negative impact is consistent with the results that the increasing extent of neck dissection was associated with worse OS, DSS and PFS in the univariate analysis. For patients with a single positive cervical node without extranodal extension, neck dissection of levels II-IV, including at least 18 identified lymph nodes, without adjuvant radiotherapy has been shown to be a possible therapeutic regime if the patient’s compliance allows a thorough follow-up [16]. 

In contrast, in advanced-stage HNCUPs requiring extended neck dissection, the likelihood of trimodal therapy being necessary increases in order to achieve an optimal oncologic outcome [16]. In this regard, several retrospective studies demonstrated the advantage of locoregional control for primary surgical therapy followed by adjuvant chemoradiation therapy [25,26,27]. Accordingly, a combined approach to achieve the best possible oncological outcomes for the patient should be recommended.

Concomitant cisplatin-based chemotherapy in addition to adjuvant radiotherapy provides a survival benefit for patients with head and neck squamous cell carcinoma with positive resection margins and ENE [28,29]. However, apart from these known data in HNSCC, there are no prospective studies on this topic with regard to HNCUP. Nevertheless, retrospective studies reported benefits for adjuvant or primary chemoradiation [16,30,31,32,33,34]. Argiris et al. demonstrated a 5-year OS of 75% in a retrospective analysis from 1991 to 2000 with a total of 25 patients, treated preferably with adjuvant chemoradiation (22 of 25 patients) [30]. Eldeeb et al. reported a 5-year OS of 67.5% in their cohort of 40 patients evaluated retrospectively from 2000 to 2010, of whom 30 patients received adjuvant chemoradiation [32]. However, both studies lacked a comparison group and information on the HPV status.

HPV-positive oropharyngeal squamous cell carcinomas are known for their favourable prognosis in comparison to their HPV-negative counterparts, which has already led to discussion on de-escalation of therapy in the form of clinical studies [35,36,37]. However, it is controversial whether these results can be readily applied to HNCUP. In a retrospective study, Sivars et al. showed a significantly better 5-year OS for HPV-positive CUPs in comparison to HPV-negative CUPs (80% vs. 36.7%, *p* = 0.004) [37]. Axelsson et al. confirmed these results in the review of the Swedish Cancer Registry with a total of 68 patients, which are also consistent with the analysis of Cheraghlou et al. of the National Cancer Database of the United states with a total of 978 patients [8,38,39]. In our cohort, we were not able to demonstrate any advantage in the multiple analysis either for HPV-positive or -negative HNCUPs. A more detailed analysis of our patients’ history of smoking revealed that 18 out of 21 patients (85.7%) of the HPV-positive patients were smokers. This is in line with the data of Tribius et al., who identified smoking as a negative prognostic factor for HPV-positive patients [40].

Both analyses showed that distant metastases could be identified as a negative factor for the oncological outcome of the patients and thus represent a major risk factor. In addition, the presence of ENE was identified as a further negative factor in the multiple analysis. Accordingly, our study is in line with the retrospective evaluation of 58 patients by Rödel et al., who showed that ENE was a predictor for survival and that distant metastases were the most frequent cause of tumour-related death in HNCUP [23]. Furthermore, a higher LNR was associated with a significantly negative impact on DSS and PFS. Park et al. concluded similarly that a higher LNR was an independent prognostic factor for DSS based on their retrospective evaluation of 39 patients, a study that focused only on the nodal characteristics and the effect of volumetric measurements of metastatic cervical lymph nodes [41]. These results are consistent with reports of oral cavity carcinoma [42,43].

The limitations of our study are due to the inevitable bias caused by its retrospective study design. However, the results derive from a homogeneous and large patient cohort treated at a single tertiary referral cancer centre. Nevertheless, the HPV status was determined only by the p16 status and not additionally by HPV DNA. As all included patients received neck dissection followed by adjuvant therapy, the results cannot be extrapolated to patients who received one single therapy modality, i.e., surgery and radiation alone, as this was beyond the scope of this study.

As HNCUP cases are rare, studies with large numbers of patients are lacking, which makes prospective investigation challenging. In addition, existing studies often show inhomogeneous patient collectives with various applied therapies. Nevertheless, this crucial topic warrants further investigation preferably in the form of multicentre, prospective studies to increase the number of included patients. 

## 5. Conclusions

The development of distant metastases represents the major risk factor for patients with HNCUP. Furthermore, the combined approach of neck dissection followed by chemoradiation has a distinct advantage for the oncologic outcome. In addition, extranodal extension and a higher lymph node ratio are negative predictors for the oncological outcome. Therefore, multimodality therapies can be recommended to achieve optimal outcomes in HNCUP.

## Figures and Tables

**Figure 1 jcm-11-02689-f001:**
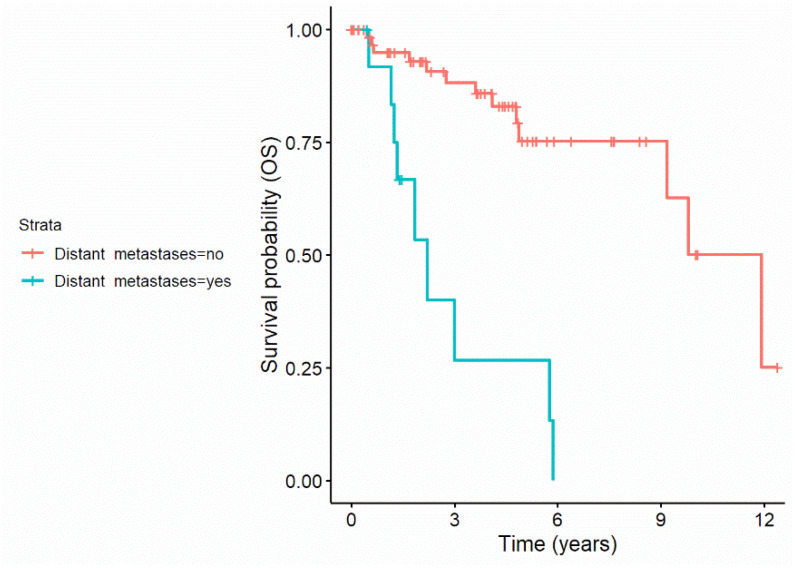
Survival curves of patients with and without distant metastases.

**Figure 2 jcm-11-02689-f002:**
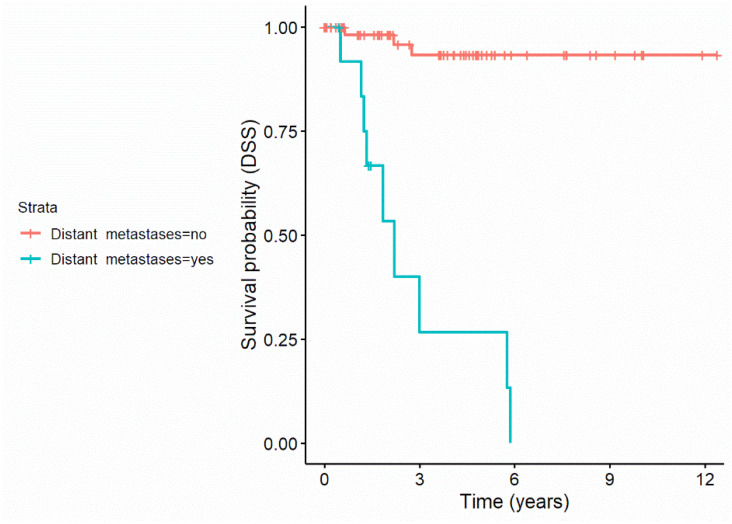
Disease specific survival curves of patients with and without distant metastases.

**Figure 3 jcm-11-02689-f003:**
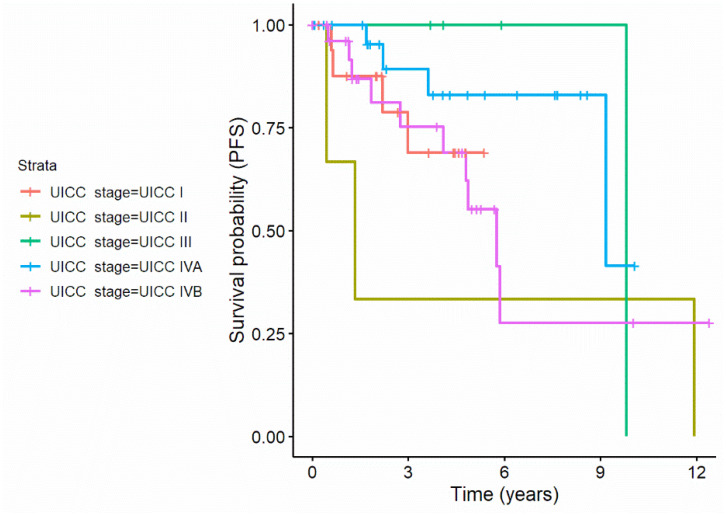
Disease specific survival curves of patients with different UICC stages.

**Table 1 jcm-11-02689-t001:** Patient characteristics.

	All Patients (n = 80)
Gender (n, %)	
Male	66/80 (82.5%)
Female	14/80 (17.5%)
Age (mean years ± SD)	60.6 ± 10.3
Nodal stage (n, %)	Total (N = 80/100%)
p16 status +/− (n, %)	59 (73.8%)/21 (26.2 %)
pN1	23 (28.7%)
p16+/−	5 (8.5%)/18 (85.7%)
pN2	3 (3.7%)
p16+/−	0 (0%)/3 (14.3%)
pN2a	5 (6.3%)
p16+/−	5 (8.5%)/0 (0%)
pN2b	9 (11.3%)
p16+/−	9 (15.3%)/0 (0%)
pN3b	40 (50.0%)
p16+/−	40 (67.7%)/0 (0%)
UICC stage (n, %)	
I	18/80 (22.5%)
II	3/80 (3.7%)
II	4/80 (5.0%)
IVA	27/80 (33.8%)
IVB	28/80 (35.0%)
Extranodal extension (n, %)	
Yes	55/80 (68.8%)
No	25/80 (31.3%)
Noxious agents	
Smoking	55/80 (68.8%)
Alcohol consumption	60/75 (80.0%)
ASA score	
1	6/80 (7.5%)
2	61/80 (76.3%)
3	13/80 (16.2%)

Abbreviations: SD = standard deviation; p16+ = association with HPV; p16− = no association with HPV; UICC = International Union Against Cancer; ASA = American Society of Anesthesiologists.

**Table 2 jcm-11-02689-t002:** Treatment characteristics.

	All Patients
Time span	
Neck dissection—adjuvant therapy (mean d ± SD)	54.3 ± 21.7
Surgical treatment modality (n, %)	
Selective neck dissection	43/80 (53.8%)
Modified radical neck dissection	14/80 (17.5%)
Radical neck dissection	23/80 (28.7%)
Adjuvant treatment modality (n, %)	
Radiation therapy	8/80 (10.0%)
Chemoradiation therapy	72/80 (90.0%)
Radiation dose in Gy (mean ± SD)	70.05 ± 8.26
Chemotherapy (n, %)	
Cisplatin/carboplatin + 5-FU	69/72 (95.8%)
Other	3/72 (4.2%)
Number of removed lymph nodes	20.8 ± 10.5
Lymph node ratio (LNR; mean ± SD)	0.20 ± 0.22

Abbreviations: SD = standard deviation; Gy = Gray; 5-FU = 5-Fluorouracil.

**Table 3 jcm-11-02689-t003:** Oncological outcomes.

	All Patients (n = 80)
Recurrence of disease:	
Distant metastases	6 (7.5%)
locoregional recurrence	13 (16.3%)
5-year OS rate	
Events (death)	22 (27.5%)
KM estimate (95% CI)	67.7% (54.2–81.2%)
5-year DSS rate	
Events (death)	12 (15.0%)
KM estimate (95% CI)	82.3 % (72.1–92.5%)
5-year PFS rate	
Events (death)	19 (23.8%)
KM estimate (95% CI)	72.8% (61.8–83.8%)

Abbreviations: OS = overall survival; DSS = disease-specific survival; PFS = progression-free survival; KM = Kaplan–Meier.

**Table 4 jcm-11-02689-t004:** Univariate and multiple Cox regression for OS.

	Univariate Cox Regression			Multiple Cox Regression		
	HR (95%-CI)	Wald Statistic	N (%)	HR (95%-CI)	Wald Statistic	
**Age**	1.05 (0.99–1.09)	*p* = 0.063	80			
**Nodal stage (pN)**						
pN1 (REF)			23 (28.8%)			
pN2	2.73 (0.26–29.01)	*p* = 0.406	3 (3.8%)			
pN2a	0 (0–) *	*p* = 0.985	5 (6.3%)			
pN2b	0.85 (0.09–8.21)	*p* = 0.891	9 (11.3%)			
pN3b	2.91 (0.83–10.17)	*p* = 0.095	40 (50.0%)			
**UICC-stage**						
UICC 1 (REF)			18 (22.5%)			
UICC 2	2.12 (0.21–21.95)	*p* = 0.528	3 (3.8%)			
UICC 3	0 (0–) *	*p* = 0.985	4 (5.0%)			
UICC IVa	0.42 (0.07–2.51)	*p* = 0.342	15 (18.8%)			
UICC IVb	2.87 (0.81–10.14)	*p* = 0.102	40 (50.0%)			
ENE(+)	1.87 (0.68–5.13)	*p* = 0.226	55 (68.8%)			
**Surgical treatment modality**						
Selective neck dissection (REF)			43 (53.8%)			
**Modified-radical neck dissection**	**3.77 (1.25–11.40)**	***p* = 0.019**	14 (17.5%)			
Radical neck dissection	2.24 (0.83–6.04)	*p* = 0.110	23 (28.8%)			
**Radiotherapy vs. chemoradiotherapy**	**3.54 (1.16–10.81)**	***p* = 0.027**	7 (8.8%)	**4.45 (1.40–14.17)**	***p* = 0.012**	7 (8.8%)
No. of removed lymph nodes	0.98 (0.93–1.02)	*p* = 0.303	80			
LNR	5.01 (0.83–30.31)	*p* = 0.080	80			
Locoregional metastases (yes)	2.70 (0.61–11.85)	*p* = 0.189	6 (7.5%)			
**Distant metastases (yes)**	**7.41 (2.95–18.62)**	***p* < 0.001**	13 (16.3%)	**8.24 (3.21–21.15)**	***p* < 0.001**	13 (16.3%)
p16 status (positive)	1.70 (0.67–4.29)	*p* = 0.261	21 (26.3%)			
Time span ND—adjuvant therapy	1.00 (0.98–1.02)	*p* = 0.999	80			

Abbreviations: HR = hazard ratio; REF = Reference; LNR = lymph node ratio; UICC = International Union Against Cancer; ENE = extranodal extension; ND = neck dissection. * 95%-CI could not be estimated.

**Table 5 jcm-11-02689-t005:** Univariate and multiple Cox regression for DSS.

	Univariate Cox Regression			Multiple Cox Regression		
	HR (95%-CI)	Wald Statistic	N (%)	HR (95%-CI)	Wald Statistic	N (%)
**Age**	0.99 (0.92-1.05)	*p* = 0.646	80			
**Nodal stage (pN)**						
pN1 (REF)			23 (28.8%)			
pN2	2.52 (0.25–25.58)	*p* = 0.433	3 (3.8%)			
pN2a	0 (0–) *	*p* = 0.988	5 (6.3%)			
pN2b	0 (0–) *	*p* = 0.981	9 (11.3%)			
pN3b	1.68 (0.44–6.34)	*p* = 0.446	40 (50.0%)			
**UICC stage**						
UICC 1 (REF)			18 (22.5%)			
UICC 2	1.52 (0.14–16.28)	*p* = 0.778	3 (3.8%)			
UICC 3	0 (0–) *	*p* = 0.986	4 (5.0%)			
UICC IVa	0.14 (0.01–1.44)	*p* = 0.098	15 (18.8%)			
UICC IVb	1.28 (0.32–5.10)	*p* = 0.729	40 (50.0%)			
ENE (+)	42.90 (0.31–5891.45)	*p* = 0.134	55 (68.8%)			
**Surgical treatment modality**						
Selective neck dissection (REF)			43 (53.8%)			
**Modified radical neck dissection**	**5.35 (1.06–27.09)**	***p* = 0.043**	14 (17.5%)			
**Radical neck dissection**	**4.52 (1.23–18.14)**	***p* = 0.033**	23 (28.8%)			
Radio- vs. chemoradiotherapy	1.59 (0.21–12.36)	*p* = 0.657	7 (8.8%)			
No. of removed lymph nodes	0.97 (0.91–1.03)	*p* = 0.369	80			
**LNR**	**12.93 (1.54–108.45)**	***p* = 0.018**	80			
**Recurrent locoregional disease (yes)**	**4.89 (1.03–23.15)**	***p* = 0.046**	6 (7.5%)			
**Distant metastases (yes)**	**23.79 (6.32–89.56)**	***p* < 0.001**	13 (16.3%)	**23.79 (6.32–89.56)**	***p* < 0.001**	13 (16.3%)
p16 status (positive)	1.86 (0.54–6.32)	*p* = 0.324	21 (26.3%)			
Time span ND—adjuvant therapy	1.001 (0.98–1.03)	*p* = 0.930	80			

Abbreviations: HR = hazard ratio; REF = Reference; LNR = lymph node ratio; UICC = International Union Against Cancer; ENE = extranodal extension; ND = neck dissection. * 95%-CI could not be estimated.

**Table 6 jcm-11-02689-t006:** Univariate and multiple Cox regression for PFS.

	Univariate Cox Regression			Multiple Cox-Regression		
	HR (95%-CI)	Wald Statistic	N (%)	HR (95%-CI)	Wald Statistic	N (%)
**Age**	0.99 (0.92–1.05)	*p* = 0.646	80			
**Nodal stage (pN)**						
pN1 (REF)			23 (28.8%)			
pN2	2.52 (0.25–25.58)	*p* = 0.433	3 (3.8%)			
pN2a	0 (0–) *	*p* = 0.988	5 (6.3%)			
pN2b	0 (0–) *	*p* = 0.981	9 (11.3%)			
pN3b	1.68 (0.44–6.34)	*p* = 0.446	40 (50.0%)			
**UICC stage**						
UICC 1 (REF)			18 (22.5%)			
UICC 2	1.52 (0.14–16.28)	*p* = 0.778	3 (3.8%)			
UICC 3	0 (0–) *	*p* = 0.986	4 (5.0%)			
UICC IVa	0.14 (0.01–1.44)	*p* = 0.098	15 (18.8%)			
UICC IVb	1.28 (0.32–5.10)	*p* = 0.729	40 (50.0%)			
ENE(+)	**8.69 (1.16–65.37)**	***p* = 0.036**		**2.10 ± 1.03**	**8.13 (1.08–61.34)**	***p* = 0.042**
**Surgical treatment modality**						
Selective neck dissection (REF)						
**Modified-radical neck dissection**	**5.42 (1.69–** **17.40)**	***p* = 0.004**				
**Radical neck dissection**	**2.51 (0.79–** **7.98)**	***p* = 0.119**				
Radio- vs. chemoradiotherapy	1.95 (0.45; 8.48)	*p* = 0.375				
No. of removed lymph nodes	1.01 (0.97; 1.06)	*p* = 0.554				
**LNR**	**7.93 (1.38–45.45)**	***p* = 0.020**		**1.82 ± 0.87**	**6.17 (1.13–33.76)**	***p* = 0.036**
p16 status (positive)	0.78 (0.26–2.34)	*p* = 0.654				
Time span ND—adjuvant therapy	1.001 (0.98–1.02)	*p* = 0.946				

Abbreviations: HR = hazard ratio; REF = Reference; LNR = lymph node ratio; UICC = International Union Against Cancer; ENE = extranodal extension; ND = neck dissection. * 95%-CI could not be estimated.

## Data Availability

The datasets used and/or analysed during the current study are available from the corresponding author on reasonable request.

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
