# Peer review of "Factors Influencing the Outcome of Head and Neck Cancer of Unknown Primary (HNCUP)"

_jcm, 2022, doi:10.3390/jcm11102689_

Round 1

Reviewer 1 Report

The paper Factors influencing the outcome of head and neck cancer of unknown primary (HNCUP) is not suitable for publication in the present form. The main problem is data analysis, with following points

1. 5-year overall survival, 5-year disease-specific survival rate 5-year progression-free survival rate are not optimal points for studies aiming at identification of prognostic /therapy goals – these parameters are more useful for quality control in large studies.

2. Univariate Cox regression is not needed. Data analysis in oncology should be exclusively multivariate [Reference Crowley et al, 2011]. Also, the combination of univariate and multivariate methods makes tables hard to follow (Tables 4 and 5).

3. Optimal way to analyze presented data is not Cox regression - recursive partitioning should be used instead [Reference Crowley et al, 2012]. This method uses multivariate Cox regression in a hidden way, but the results are presented as survival tree, which is usually easy to interpret. Survival tree identifies prognostic subgroups, and explains influence of variables on survival. Data are already formatted for analysis and an efficient data analyst needs only a few hours of work to re-analyze data. In this way paper will be more concise, more clear and easier to follow for readers. The graphical nature of survival tree is also for benefit of less numerically inclined readers.

Reference:

Crowley, J et al. Handbook of Statistics in Clinical Oncology. 3Rd ed. Chapman and Hall/CRC , Boca Raton, 2012.

Author Response

Manuscript: jcm-1653930

Dear Editor,

Thank you very much for the opportunity to revise our manuscript "Factors influencing the outcome of head and neck cancer of unknown primary (HNCUP)".

As requested, we have written a point-by-point response letter and have thoroughly addressed each comment by the editor.

Our responses are written in italics below the respective comment. The changes in the script have been marked in red.

We very much appreciate the chance to significantly improve our initial manuscript through the reviewer’s and editor’s comments and hope that the revised manuscript is now suitable for publication.

Best regards

  1. Balk

Reviewer 1:

The paper Factors influencing the outcome of head and neck cancer of unknown primary (HNCUP) is not suitable for publication in the present form. The main problem is data analysis, with following points

  1. 5-year overall survival, 5-year disease-specific survival rate 5-year progression-free survival rate are not optimal points for studies aiming at identification of prognostic /therapy goals – these parameters are more useful for quality control in large studies.

We apologize, there was a mistake in reporting the results. Although we do report the 5year-Survival in this study, risk factors were not calculated for the 5 year survival but rather for the whole observation time.This has been corrected in the manuscript.

  1. Univariate Cox regression is not needed. Data analysis in oncology should be exclusively multivariate [Reference Crowley et al, 2011]. Also, the combination of univariate and multivariate methods makes tables hard to follow (Tables 4 and 5).

We conducted univariate cox regression first, due to the small sample size. By this, we hoped to reduce the number of predictores for the final multivariate regression, to avoid overfitting the model.Therefore, to enease understanding, the table is displayed now in a slightly changed design.

  1. Optimal way to analyze presented data is not Cox regression - recursive partitioning should be used instead [Reference Crowley et al, 2012]. This method uses multivariate Cox regression in a hidden way, but the results are presented as survival tree, which is usually easy to interpret. Survival tree identifies prognostic subgroups, and explains influence of variables on survival. Data are already formatted for analysis and an efficient data analyst needs only a few hours of work to re-analyze data. In this way paper will be more concise, more clear and easier to follow for readers. The graphical nature of survival tree is also for benefit of less numerically inclined readers.

Answer to comment 2 and 3:

Thank you for your recommendations. We conducted univariate cox regression first, due to the small sample size. By this we hoped to reduce the number of predictores for the final multivariate regression, to avoid overfitting the model. Also we feared, that purely multivariate analyses could lead to the discounting of some predictors which would otherwise be marked for further studies.

Still, we agree that a tree-based analysis does benefit the study and we added it accordingly. It has to be noted though, that small sample size is an issue in this analyses, too, as can be seen by the increase of cross validation error as we add more and more splits to the model. We would thus argue in favor of keeping both analyses to gain a maximum confidence in the results, which overlap substantially.

We have submitted the detailled results of the tree-based analysis as a seperate file, in case you would like to have more in depth information. Please let us know, if you would like to have further changes.

To facilitate understanding, the table of cox regression is displayed now in a slightly changed design.

Reviewer 2 Report

This is a retrospective review of the treatment outcomes of a group of patient with HNSCC with unknow primary in a single center.   The author review the cases in his center and systemically analyse the data with appropriate statistic. He concluded that  adjuvant treatment with radiation without concomitant chemotherapy, as well as the presence of distant metastases are, among others, significant predictors for the overall survival of HNCUP patient.  

I agreed with his conclusion base on his patient cohort.  The inferior result of surgery + radiotherapy alone may be biased by a large number of N3b(50%) disease in this cohort.  

In summary, HNSCC of unknow primary is not a common disease, this paper summarize a cohort of 80 patients in a single center for over 13 years.  It did not give any new information or new idea but it did contribute a well written retrospective analysis for this disease entity.

Author Response

Manuscript: jcm-1653930

Dear Editor,

Thank you very much for the opportunity to revise our manuscript "Factors influencing the outcome of head and neck cancer of unknown primary (HNCUP)".

As requested, we have written a point-by-point response letter and have thoroughly addressed each comment by the editor.

Our responses are written in italics below the respective comment. The changes in the script have been marked in red.

We very much appreciate the chance to significantly improve our initial manuscript through the reviewer’s and editor’s comments and hope that the revised manuscript is now suitable for publication.

Best regards

  1. Balk

Reviewer 2:

This is a retrospective review of the treatment outcomes of a group of patient with HNSCC with unknow primary in a single center.   The author review the cases in his center and systemically analyse the data with appropriate statistic. He concluded that  adjuvant treatment with radiation without concomitant chemotherapy, as well as the presence of distant metastases are, among others, significant predictors for the overall survival of HNCUP patient.  

I agreed with his conclusion base on his patient cohort.  The inferior result of surgery + radiotherapy alone may be biased by a large number of N3b(50%) disease in this cohort.  

In summary, HNSCC of unknow primary is not a common disease, this paper summarize a cohort of 80 patients in a single center for over 13 years.  It did not give any new information or new idea but it did contribute a well written retrospective analysis for this disease entity.

We are very thankful for this comment and hope that the revised manuscript is now suitable for publication.